# Top predators constrain mesopredator distributions

Thomas M. Newsome[1,2,3,4], Aaron C. Greenville[2,5], Duško Ćirović[6], Christopher R. Dickman[2,5], Chris N. Johnson[7], Miha Krofel[8], Mike Letnic[9], William J. Ripple[3], Euan G. Ritchie[1], Stoyan Stoyanov[10] & Aaron J. Wirsing[4]

Top predators can suppress mesopredators by killing them, competing for resources and instilling fear, but it is unclear how suppression of mesopredators varies with the distribution and abundance of top predators at large spatial scales and among different ecological contexts. We suggest that suppression of mesopredators will be strongest where top predators occur at high densities over large areas. These conditions are more likely to occur in the core than on the margins of top predator ranges. We propose the Enemy Constraint Hypothesis, which predicts weakened top-down effects on mesopredators towards the edge of top predators' ranges. Using bounty data from North America, Europe and Australia we show that the effects of top predators on mesopredators increase from the margin towards the core of their ranges, as predicted. Continuing global contraction of top predator ranges could promote further release of mesopredator populations, altering ecosystem structure and contributing to biodiversity loss.

[1] School of Life and Environmental Sciences, Centre for Integrative Ecology, Deakin University, Geelong, Victoria 3125, Australia. [2] School of Life and Environmental Sciences, The University of Sydney, Sydney, New South Wales 2006, Australia. [3] Global Trophic Cascades Program, Department of Forest Ecosystems and Society, Oregon State University, Corvallis, Oregon 97331, USA. [4] School of Environmental and Forest Sciences, University of Washington, Seattle, Washington 98195, USA. [5] National Environmental Science Programme Threatened Species Recovery Hub, University of Sydney, Sydney, New South Wales 2006, Australia. [6] Faculty of Biology, University of Belgrade, Belgrade 11000, Serbia. [7] School of Biological Sciences and Australian Research Council Centre of Excellence for Australian Biodiversity and Heritage, University of Tasmania, Hobart, Tasmania 7001, Australia. [8] Wildlife Ecology Research Group, Department of Forestry, Biotechnical Faculty, University of Ljubljana, Ljubljana 1000, Slovenia. [9] Centre for Ecosystem Science, and School of Biological, Earth and Environmental Sciences, University of New South Wales, Sydney, New South Wales 2052, Australia. [10] Wildlife Management Department, University of Forestry, Sofia 1756, Bulgaria. Correspondence and requests for materials should be addressed to T.M.N. (email: tnew5216@uni.sydney.edu.au).

A key goal of ecology is to understand the factors that shape species' distributional limits, which to date have been examined largely in relation to abiotic drivers such as climate[1]. The role of biotic interactions, such as predation and competition, in determining range boundaries remains poorly understood[2,3], even though such interactions can have strong effects[4]. Accordingly, there is a need to examine how biotic factors limit species' distributions, especially across a range of habitats that have different levels of abiotic stressors[3]. Such assessments are required to predict species' assemblages in the face of ongoing global environmental disturbance associated with habitat loss and modification, biological invasions, decline of apex consumers and climate change[4–6].

Interspecific competition is often especially strong among predators[7]. Negative relationships between the local abundances of top predators and mesopredators have been documented in many cases[8]. If this pattern scales up, mesopredator abundance should vary with spatial variation in the abundance of top predators. Ecological theory predicts that populations at the periphery of their geographic ranges will have low densities, whereas more centrally located populations will have higher densities[9,10]. Therefore, suppression of mesopredators may be greatest well within a top predator's range where the abundances of that predator are highest. In contrast, for some distance within the edge of the top predator's range, suppression of mesopredators may occur but be insufficient to drive mesopredator abundances close to zero. These effects have the potential to influence entire ecological communities[11,12], but there have been few quantitative efforts[7,13–16] to test whether suppression of mesopredators varies according to the distribution and abundance of top predators at large spatial scales. Moreover, nothing is known about how suppression might vary on the edge of top predator ranges or across different regions and ecological contexts.

We tested whether mesopredator abundance is affected by the spatial distribution and abundance of top predators across extensive landscapes. We propose the Enemy Constraint Hypothesis (ECH), which predicts relatively weak top-down control of mesopredators on the edge of top predator ranges, a progressive decline in mesopredator abundance with increasing distance into the core of top predator ranges, and mesopredator numbers approaching zero where top predator abundance is at a peak (Fig. 1). We tested the ECH by analysing bounty data from North America (Saskatchewan), Europe (Bulgaria/Serbia) and two regions from Australia in the State of Queensland (referred to as Australia East and Australia West). Predator distributions in these study areas provide opportunities to explore theoretical questions under a natural experimental framework[13]. In North America and Europe, grey wolves *Canis lupus* (top predator) were extirpated by humans from parts of their historical range, resulting in the formation of new range boundaries (Fig. 2). A similar process occurred for the dingo *Canis dingo* (top predator) in Australia (Fig. 2). We used the existence of these new range boundaries to quantify changes in mesopredator abundance on either side of the range edge. The mesopredators include the coyote *Canis latrans* (North America), golden jackal *Canis aureus* (Europe) and red fox *Vulpes vulpes* (Australia).

Our results, consistent across three continents, suggest that top predators can suppress mesopredators to the point of complete exclusion, but only when top predators occur at high densities over large areas. The results suggest further that these conditions are more likely to occur at the core than on the margins of top predator ranges, providing support for the ECH. The results have important implications for understanding species interactions and niches, as well as the ecological role of top predators. More broadly, there is a need to determine the causal mechanisms that

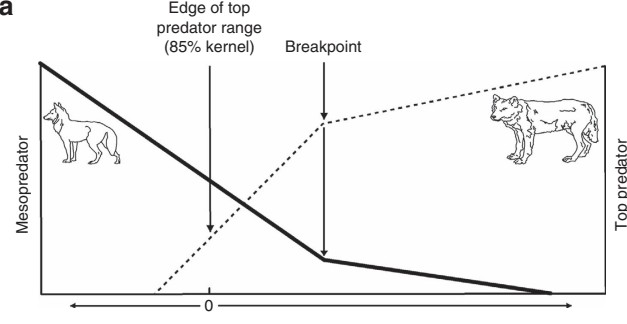

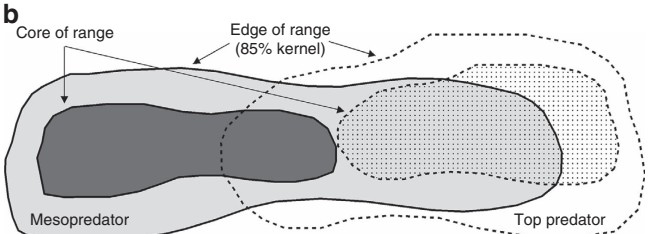

**Figure 1 | Conceptual model of the Enemy Constraint Hypothesis using top predators and mesopredators as the subjects.** (**a**) On the edge of a top predator's range, mesopredator abundance should decline as top predator abundance increases. The breakpoint for the mesopredator indicates where their abundance starts to become close to zero. The breakpoint for the top predator indicates where their abundance starts to decline sharply on the edge of the range. A breakpoint is not necessary for the ECH to hold, but it may be indicative of a key threshold where there is a sharp change in top predator or mesopredator abundance, and is therefore useful to assess. (**b**) The relationship in **a** should manifest where mesopredators overlap spatially with the edge of a top predator's range, with the relationship potentially applying more widely to other predator dyads that strongly interact and compete for similar resources, or even to any strongly interacting competitive species dyads ('enemies') including relationships involving parasites or pathogens.

drive the observed trends (for example, predation, competition or a mixture of both), and whether the results of the ECH apply to other predator dyads that strongly interact and compete for similar resources, or even to any strongly interacting competitive species dyads (which we term 'enemies', Fig. 1).

## Results

**Indices of abundance.** The range limits for the species considered in the study are shown in Figs 2 and 3. As expected, indices of abundance based on bounty returns for each top predator were low on the edge of its range and increased towards its range core (Figs 3 and 4). Mesopredator abundance indices were higher outside the current ranges of top predators and declined progressively with distance from the edge into each top predator's range (Figs 3 and 4).

**Breakpoints.** In North America, Europe and Australia West, abundance indices of mesopredators were close to zero within each top predator's range as indicated by breakpoints at 384, 214 and 320 km from the range edge, respectively (Fig. 4, Supplementary Table 1,2). Breakpoints in the abundance indices of top predators in North America, Europe, Australia West and Australia East occurred at 241, 208, 259 and 302 km from the range edge, respectively (Fig. 4, Supplementary Table 1,2). There was no clear breakpoint where mesopredator abundance indices in Australia East were close to zero, although the shape of the plot was similar to all other sites (Fig. 4, Supplementary Table 1,2).

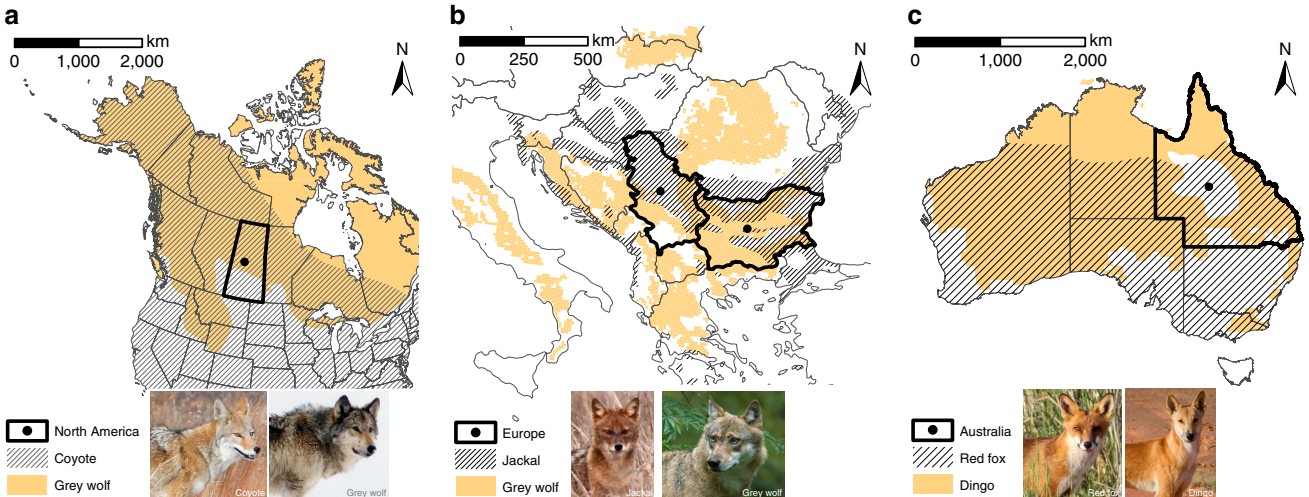

**Figure 2 | Predator distribution during the study periods in each continent.** Distribution is shown for (**a**) coyotes (hashed) and grey wolves (orange) in North America (Saskatchewan)[7], (**b**) golden jackals (hashed) and grey wolves (orange) in Europe (Bulgaria and Serbia)[16,19,27] and (**c**) red foxes (hashed) and dingoes (orange) in Australia (Queensland)[13,21]. Note that the scales differ between continents. The black outline with dot in the centre denotes the study region in each continent, with Bulgaria (right) and Serbia (left) shown separately in **b**.

**Spatial correlation.** In North America (both species), Australia East (both species) and Australia West (top predator) there was no major spatial correlation based on plots of residuals versus their spatial co-ordinates (Supplementary Figs 1,2,5,6,7), as indicated by the lack of a pattern whereby groups of positive or negative residuals are spatially clumped close to each other[17]. In Europe (both species) and Australia West (mesopredator) there was minor clumping of the positive and negative residuals, although not in any particular direction (Supplementary Figs 3,4,8).

## Discussion

The observed declines in indices of mesopredator abundance could have been due to environmental gradients[15], land use changes[18] or other abiotic stressors[3] that made conditions progressively less suitable for each mesopredator. However, the mesopredators we studied are habitat generalists; the coyote occurs in a range of environments including urban areas and as far north as Alaska[7], while the golden jackal occurs as far north as Estonia and as far west as Switzerland[19]. Accordingly, the environmental conditions within the core ranges of our focal top predators are suitable for these mesopredators, leading us to expect that they would have occupied larger areas in the absence of the top predator. Furthermore, we observed similar patterns of abundance indices of the red fox in two distinctly different physical environments. Australia West is predominantly arid, whereas Australia East is more productive and contains structurally complex forest areas. Yet, in both cases abundance indices of red foxes declined progressively within the range of the dingo.

An alternative explanation is that top predators exert negative effects on mesopredators at all densities throughout their ranges, but mesopredator numbers dwindle from the edge to the centre of the top predator range because they are progressively cut off from their larger source populations. This scenario would represent a 'rescue effect'[20], by which small and isolated mesopredator populations deep within the ranges of top predators are prevented from going extinct by continuing inputs of immigrants. However, mesopredator abundance indices declined close to zero within top predator's ranges in all cases assessed, therefore showing that any immigration, progressively, became ineffective (Fig. 4). Thus,

while the 'rescue effect' may have contributed to the large distances that mesopredators occurred within the ranges of top predators, no mesopredator is likely to show such large movements or range sizes that it would fully explain the >200 km breakpoints.

The use of bounty data could have confounded the results if the number of predators killed was influenced by (i) bounty price/human effort, (ii) background fluctuations in populations or (iii) poor weather for trapping and hunting. However, the same bounty price was paid for a given predator in each hunting unit, so bounty prices are unlikely to have driven changes in human effort so as to produce the spatial gradients in bounty returns that we observed. All the other factors apply equally to top predators and mesopredators because of their biological similarities, so they also are not likely to have driven the observed spatial patterns. The bounty data we used are from published studies[7,13,16], and bounty data are commonly used to derive indices of predator abundance at large spatial scales[15]. We are therefore confident that the bounty data reflect spatial variation in predator abundances. This argument is strengthened by the consistent results we found across three separate continents, all of which have different abiotic stressors, using different predator pairs. Furthermore, despite the bounty data from Australia being collected much earlier (1950s) in comparison to that in North America (1982–2011) and Europe (2000–2009), the results in Australia are corroborated by more recent evidence showing that dingoes can suppress red fox populations[21].

In the absence of other available data, we suggest that top predators progressively exert more top-down pressure the more abundant they become towards the core of their ranges, such that mesopredators disappear when deaths (induced by top predator competition or killing) exceed births. The spatial gradient across the range edge of the top predators that we examined is essentially a surrogate for top predator abundance. Although not essential for supporting the ECH, the existence of breakpoints in the fitted lines for mesopredators and top predators may identify abundance thresholds at which the top predator becomes ecologically effective[22] at suppressing the mesopredator, or the key threshold beyond which the ecological effectiveness of the top predator increases rapidly (Fig. 4). By implication, relationships between top predators and mesopredators at large spatial scales are frequency dependent[23], with top predators exerting

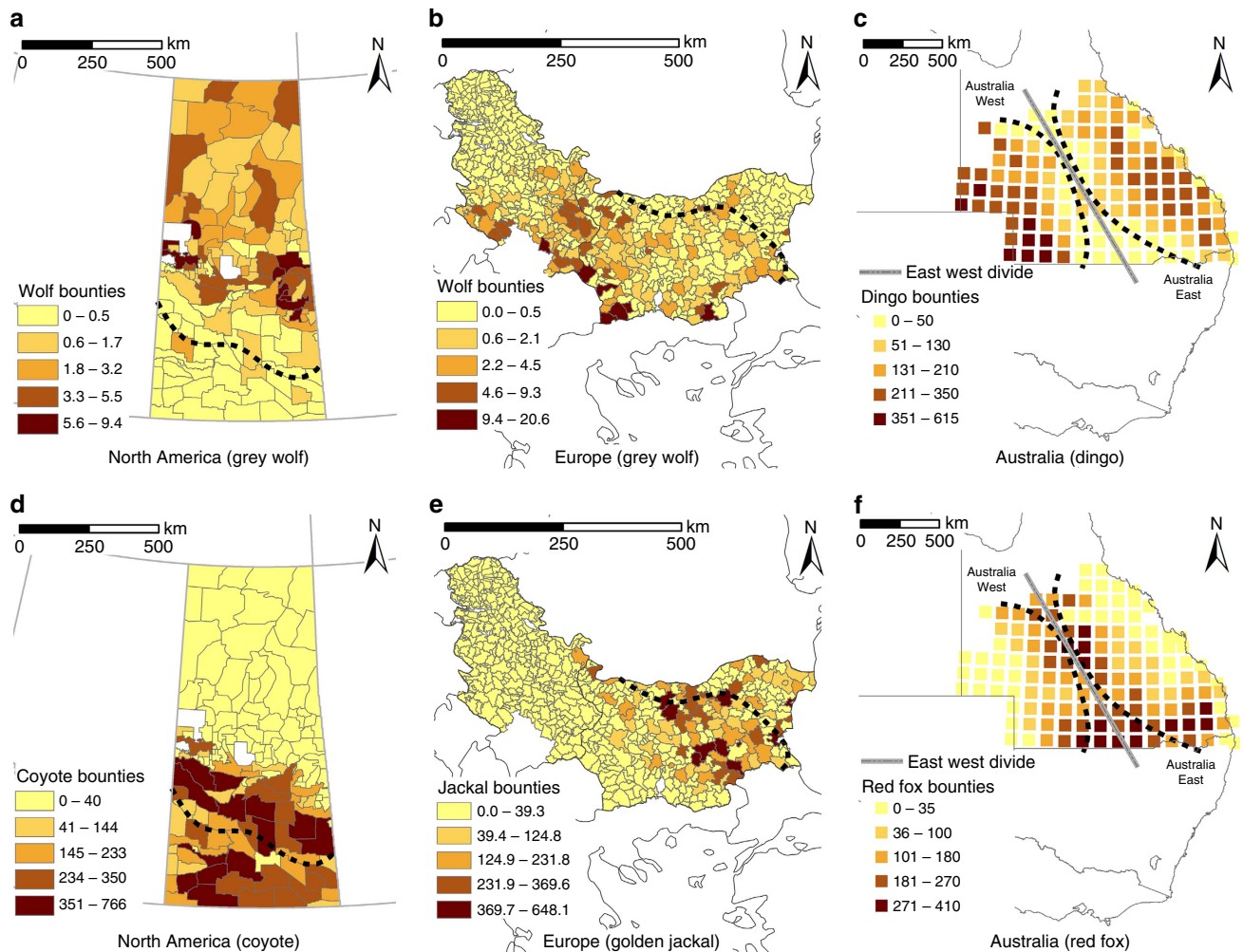

**Figure 3 | Predator bounties and top predator range edges in each continent.** The number of bounties (representing the number of animals killed) are given for each hunting unit in North America (collated from 1982 to 2011) and Europe (collated from 2000 to 2009), whereas each square in Australia represents the number of bounties in a 100 × 100 km area (collated from 1951 to 1952). Hunting units with no bounty data were excluded from the analysis. Top predators are in **a**–**c**. Mesopredators are in **d**–**f**. Darker colours within each hunting unit indicate greater bounty return numbers and by inference, a higher abundance for the respective predator. Range edges (dashed black lines) are 85% kernel density probability contours based on the number of top predator bounties. Australia was divided into two sections for the analysis (east and west) as shown. Note that the scales differ between continents.

disproportionately higher levels of mesopredator suppression as their abundance increases.

Our analysis supports historical accounts linking the rapid expansion of mesopredator populations to the extirpation of top predators[24], and suggests further that top predators can suppress mesopredator populations, even to the point of complete exclusion, as demonstrated in smaller scale studies[25]. However, the mere presence of a top predator may not be sufficient to exert strong suppressive effects on mesopredators. This observation could explain why some studies have documented only weak effects of top predators on mesopredators[26]. Furthermore, the mesopredator breakpoints identified in North America and Australia West were 143 and 61 km away from the top predator breakpoints respectively. Both these mesopredator breakpoints occurred well into each top predator's range suggesting there are expansive areas where these predators coexist (Fig. 4). In Europe and Australia East the top predator abundance indices also decreased at distances well away from the range edge (Fig. 4). These decreases did not correspond with an increase in mesopredator abundance indices in either case, indicating the presence of abiotic stressors or that the habitats are not well suited for either species. In the case of the latter, the

bounty data suggest that both grey wolves and golden jackals are virtually absent from northern Serbia where there is intensive agriculture, a finding that supports other studies[27,28]. Similarly, Eastern Australia (especially along the coastline) is a heavily human-modified system in comparison to inland Australia, and so this may explain the decline in dingo abundance indices that we found on the far eastern side there.

Another factor that could limit top-down suppression of mesopredators is that the social stability of top predators is often altered by anthropogenic control[29,30], such that human influences dampen the strength of top-down forcing[31,32] and lead to a shift in ecological state to a bottom-up driven system with increased mesopredators[31]. In our case studies, the ranges of top predators contracted due to killing by humans and human modifications to the environment (for example, habitat loss and fragmentation). When assessing the ability of top predators to suppress mesopredators, it may therefore be necessary to consider social stability of top predators and other anthropogenically driven influences on landscapes and foodwebs[18]. Such investigations would help to ascertain the circumstances where top predators and mesopredators coexist, or where suppression occurs versus complete exclusion. When considering grey wolves and coyotes,

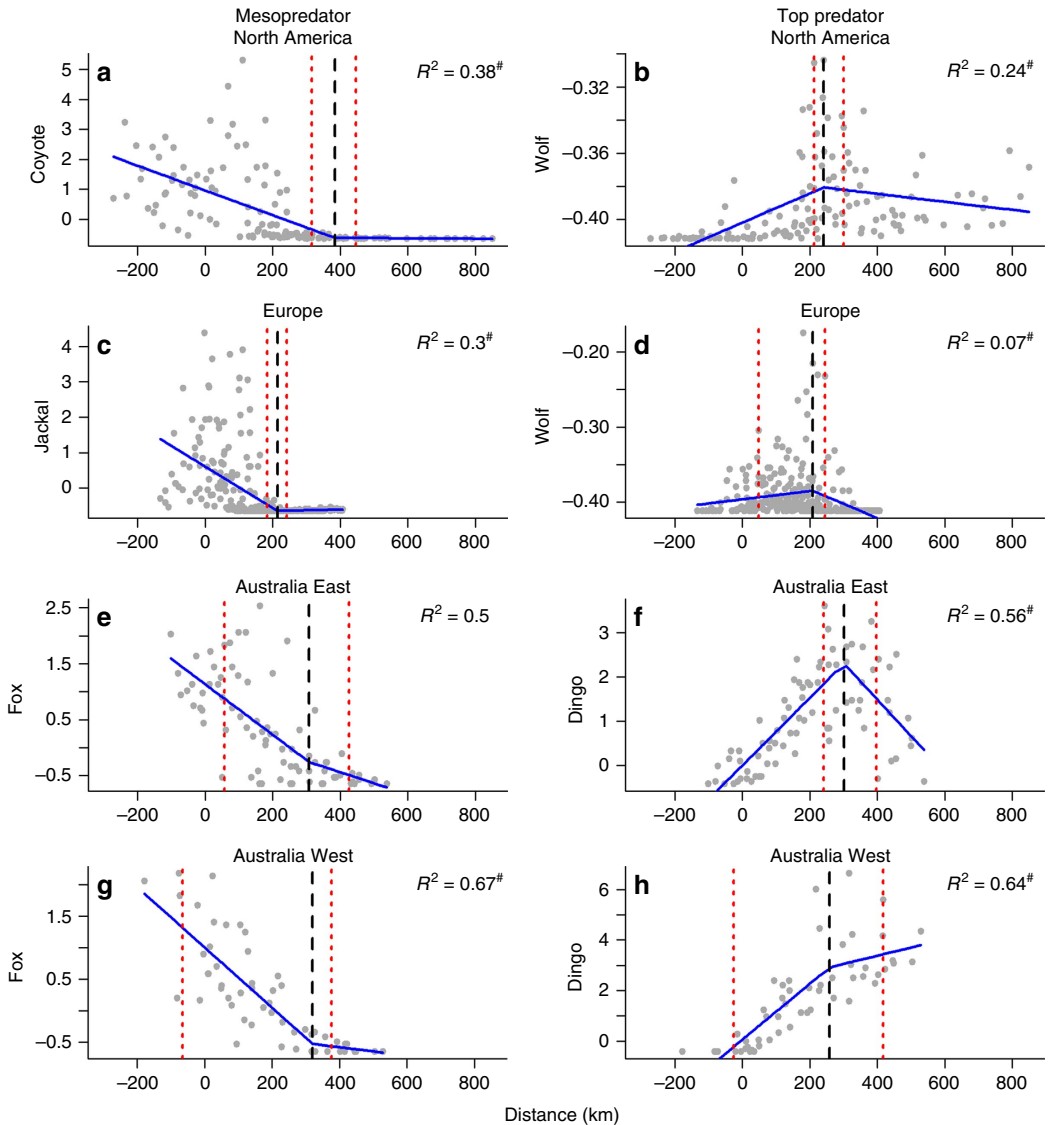

**Figure 4 | Relationships between top predator and mesopredator abundance indices and distance to the edge of top predator ranges in each continent.** Distance is calculated from the centroid of each hunting unit to the edge of the top predator's range in each continent (Fig. 3). The point where there is a sharp change in slope is indicated by the dashed black line ( ± 95% confidence intervals; dashed red lines). Significant differences ($P < 0.05$) in the slopes of the regression lines are indicated by the hash above the $R^2$ value. The range edge is set at zero and based on 85% kernel density probability contours in Fig. 3. Hunting units with negative values are located outside top predator ranges. For the y axis, abundance indices (bounty values) were standardized by subtracting the mean and dividing by the s.d. (z-scores) to allow for direct comparison among continents. Top predators are in **b**, **d**, **f**, **h**. Mesopredators are in **a**, **c**, **e**, **g**.

complete mesopredator exclusion is possible, at least at historical levels of top predator abundance across large landscapes[24]. Even more recently, complete exclusion has been found in relatively closed systems (for example, Isle Royale National Park, USA[25]), although coexistence has been found in more open systems where constant immigration by the mesopredators is possible (for example, Riding Mountain National Park, Canada[33]). Our case studies suggest there is a point where mesopredators are virtually absent well within top predator ranges, but it is not possible to determine if this reflects complete exclusion or simply low detection based on bounty returns.

The general predictions of the ECH can be tested for other predator dyads that strongly interact and compete for similar resources, and our predictions may be extended even further to any strongly interacting competitive species dyads including relationships involving parasites or pathogens (Fig. 1). In our

focal systems, the distance at which edge effects became manifest was > 200 km (Fig. 4), but this distance will vary with other species and ecosystem contexts. The ECH may yield insights about early and cryptic impacts of landscape modification on top-down forcing. Indeed, conservation efforts are often initiated when species are close to extinction, rather than early on when their populations are in the initial stages of decline. However, by this stage the knock-on effects (for example, mesopredator release[12]) may have already taken place, with unknown effects on ecosystem structure and biodiversity. If there is an imperative to restore top predators, or any species that can induce cascading effects that benefit ecosystems, then we need a better understanding of the abundance and spatial extent at which these species need to occur to perform their functional ecological roles. Our analysis indicates that studies assessing the strength of top-down mesopredator control will need to consider whether the

mesopredator is located on the periphery or core of the top predator's range, and whether the top predator has reduced abundance, destabilized social structure or a sporadic distribution due to some external factor or factors. In the absence of such considerations we may underestimate the potential effects of top predators on ecological communities, thereby inhibiting top predator conservation and restoration efforts.

## Methods

**Background.** Predators are controlled by humans in many parts of the world. Where governments pay hunters a bounty for predator furs or scalps it is common practice to record the location (for example, hunting unit) where the predator was killed, and records are usually collated on an annual basis. Here, we collated bounty data from North America, Europe and Australia where mesopredators occur over large areas that also feature a gradient in top predator abundance. The data collection dates vary, and reflect the availability of bounty records for each continent. We used these datasets to test our hypotheses related to top predator and mesopredator distributions and abundances. Bounty data have been used in many previous studies to derive indices of top predator and mesopredator abundances[7,15,34], based on the notion that predator abundance generally correlates positively with the number of bounty returns[7], and that bounty data can be used to compare the abundances of top predators and mesopredators because of their biological similarities[7]. No other complementary predator abundance data exist at the spatial scales required.

**North America.** We retrieved bounty data on the number of grey wolves (top predator) and coyotes (mesopredator) killed in 136 hunting units in the province of Saskatchewan (651,900 km$^2$), Canada, between 1982 and 2011. These data were collected by the Government of Saskatchewan each year based on payments made to trappers and hunters (Supplementary Table 3). The hunting units are also referred to as wildlife management zones, and these remained constant over the study period. Over the last two centuries, widespread predator control has resulted in grey wolves being largely restricted to northern forested areas in Saskatchewan, whereas they were, and continue to be, largely absent in the agricultural and rangeland areas to the south[7]. Coyotes were restricted to central North America in the 1800s, but had dispersed as far north as Alaska by the 1930s (ref. 7). Thus, by the beginning of our sampling, coyotes were present in Saskatchewan, including in areas with and without grey wolves (Fig. 2). Previous analyses of bounty data from Saskatchewan suggest that coyotes can disperse large distances ($>200$ km) into the northern forested areas where grey wolves occur[7]. In the previous analyses a coyote-to-red fox ratio was used to explore changes in the ratio of the two species on either side of grey wolf range. However, the range of the grey wolf was based on historical maps rather than bounty data, and there was no concurrent analysis of the grey wolf and coyote bounty data like that proposed herein.

**Europe.** We retrieved bounty data based on the number of grey wolves (top predator) and golden jackals (mesopredator) killed in 255 hunting units in Bulgaria (110,994 km$^2$) between 2004 and 2009, and in 148 hunting units in neighbouring Serbia (88,361 km$^2$) between 2000 and 2008. These data were collected by the respective hunting associations in each county (Supplementary Table 3). Grey wolves were sporadically distributed or largely absent in these two countries in the 1970s, but they have since increased in numbers and dispersed into eastern Serbia and Bulgaria[16,27]. Golden jackals were restricted to two isolated populations in Bulgaria in the 1960s, but they now occupy northern and southern Bulgaria and at least in small numbers across large parts of Serbia[16,19]. Thus, by the beginning of our sampling, golden jackals were present in Bulgaria and Serbia, including in areas with and without grey wolves (Fig. 2). Previous analyses of grey wolf and golden jackal bounty data from Bulgaria and Serbia suggest there is an inverse relationship between the abundances of the two species[16]. However, the full extent to which golden jackals spatially overlap in distribution with grey wolves has not been assessed previously.

**Australia.** We retrieved bounty data on the number of dingoes (top predator) and red foxes (mesopredator) killed in the southern two thirds of Queensland, Australia (1,200,000 km$^2$) between 1951 and 1952. These data were obtained from two maps published by the Queensland Government reporting the number of dingo or red fox bounties paid. The maps included locations of bounty records for both species, with one dot representing five dingoes or five red foxes. To allow for a spatial analysis and comparison of bounty records between the two species over the same area, the number of bounties paid for each species within a 100 × 100 km area was used, following previously established protocols[13]. This approach resulted in a comparison of bounty data over 145 defined locations across the study area. Dingoes were introduced into Australia ∼4,500 years ago, and at the time of European settlement (1788) they occupied the entire State of Queensland[13,21]. However, by the 1950s (following a period of intensive control), dingoes were largely absent from central Queensland in sheep grazing areas. Red foxes were

introduced into Australia following European settlement and dispersed northward from southern Australia, eventually colonizing the southern two thirds of Queensland by the 1930s. Thus, by the beginning of our sampling, red foxes were present in Queensland, including in areas with and without dingoes (Fig. 2). As with the data from Europe, an inverse relationship between the abundances of dingoes and red foxes has been found in Queensland[13]. However, the full extent to which red foxes spatially overlap in distribution with dingoes has not been assessed previously.

**Patterns of spatial overlap.** To assess patterns of spatial overlap between the top predator and mesopredator on each continent, we first mapped the number of predator bounties retrieved from each hunting unit in Arc GIS v10.1 (Environmental Systems Research Institute Inc.: Redlands, CA, USA). To standardize the data we divided the total number of bounties by the number of years of data collection. We then characterized the distribution of the top predator in each continent by calculating a kernel density estimate from the mapped bounty data described above. For North America and Europe we used the entire mapped datasets, but because dingoes were virtually absent from the centre of the Australian study area (with two core areas of occupancy on either side) we split the data into two equal portions, one representing the eastern side, and the other the western side (Fig. 3). We chose the kernel density estimate because it provides a non-parametric method of estimating probability densities that is uninfluenced by effects of grid size and placement, and can accurately estimate the densities of any shape by superimposing a grid over the data and using information from the entire sample[35]. To calculate kernel densities, we converted the bounty data in each continent into a point file using conversion tools in ArcView v10.1, with each point given the coordinates of the centroid of each hunting unit. We then used the kernel density estimator in the Geospatial Modelling Environment[36] package to create the kernel density grid for each top predator dataset. This tool calculates kernel density estimates based on a set of input points and in this case we used the converted bounty data. The cell size for the kernel density estimate was standardized across all continents by setting the grid size at the scale of 2.5 km × 2.5 km. We used the default Gaussian (bivariate normal) kernel with the smoothed cross validation method to determine the level of smoothing because this approach does not typically overestimate space use[37].

From the kernel density grid we calculated 85% probability contours for each top predator using the isopleth command in the Geospatial Modelling Environment package. The isopleth command creates a line based on a raster dataset representing a probability surface (that is, the kernel density estimate). Isopleths represent the boundary lines that contain a specified volume of a surface. For instance, the 0.95 isopleth represents the contour line containing 95% of the volume of the surface[36]. We used the 85% contour to define the edge of each top predator's distribution and used this edge as a proxy for a range boundary. The 85% contour was considered appropriate because it excluded outliers, and probability contours above 90% provided a gross overestimate of the top predator ranges based on the known distributional limits of each species (Fig. 2). Then, to assess top predator and mesopredator distributions and abundances across the study areas, we calculated the distance (km) from the centroid of each hunting unit to the closest point along the top predator's 85% probability contour edge. We set the edge as the side of the circle where top predator densities were declining (that is, the edge of the range). Because we calculated distance from both sides of the contour edge, we multiplied the distance values from bounty units on the outside of the probability contour edge by −1. This step allowed the top predator and mesopredator data to be plotted along a continuous axis covering hunting units within and outside the top predator's probability contour edge. Thus, distance values $<0$ related to bounty units outside the contour edge and those $>0$ represented bounty units inside the contour edge.

**Predator abundance and distribution patterns.** We used a piecewise linear regression to model the relationship between the top predator and mesopredator bounty data and distance to the edge of top predators range using the software R in the package siZer 0.1-4 (ref. 38) (Supplementary Methods). The piecewise linear regression allows multiple linear models to be fitted, and where the lines meet can be used to identify breakpoints where the slope of the linear function changes. Thus, the piecewise regression was chosen to determine if there are different linear trends over different regions of the data that accrued at a breakpoint, or in other words a sudden, sharp changes in slope of the line. We used the piecewise regressions, with one breakpoint that could occur at any predator bounty value. For the analysis, we excluded data from hunting units where there were no top predators and no mesopredators. The bounty values were also standardized by subtracting the mean and dividing by the s.d. (z-scores) to allow for direct comparison among continents. Although not necessary for the ECH to hold (Fig. 1), we expected the sharp change in the mesopredator bounty data to occur where their abundance was close to zero. For the top predator we expected the sharp change to occur where their abundance starts to decline on the edge of the range. To estimate $P$ values and confidence intervals (2.5 and 97.5%) around each breakpoint, we used a bootstrap method with 1,000 replacements. To test for independence (spatial correlation), we plotted the standardized residuals against their spatial co-ordinates[17].

**Data availability.** Data for Figs 3 and 4 are available from the Dryad Digital Repository http://dx.doi.org/10.5061/dryad.h1m85. Raw data are available from the first author upon request. R code is provided in Supplementary Methods.

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

## Acknowledgements

We thank the Government of Saskatchewan in Canada, the National Hunting Association—Union of Hunters and Anglers in Bulgaria and the Executive Forestry Agency for providing hunting statistics data. The study was supported by the Ministry of Education, Science and Technological Development of Serbia (TR 31009).

## Author contributions

T.M.N. conceived the study. T.M.N. and A.C.G. performed the data analysis. T.M.N. drafted the manuscript. All authors discussed the results and provided comments on the manuscript.

## Additional information

**Competing interests:** The authors declare no competing financial interests.

