## [Peer Review File · Nature Communications]

Reviewers' Comments:

Reviewer #1 (Remarks to the Author):

This paper examines the numerical suppression of mesopredator population abundance/distribution with increasing predator abundance using observed population gradients within predator ranges. The authors compiled data from 3 different predator systems across 3 different continents to illustrate a correspondence in the observed patterns, consistent with the proposed “predator edge hypothesis”. The paper is well written and the ideas are compelling. The figures are excellent and portray the story well.

My main criticism is that the data and methods are poorly described at times. I recognize that brevity is valued in a Nature Communications paper but in the absence of sharing data and code, there are more details needed to understand what was done. The authors provided a supplement of tables/figures and this supplement could be expanded with additional descriptions if such information cannot fit in the main document. Some general comments:

- 1) The bounty data serve as the primary evidence for the hypothesis but these data are barely described in the text. Where exactly do these data come from? Citing a previous paper is not the same as citing a data source, and a citation alone is not sufficient for the reader to understand the data origin. Without any context, it is strange to see square wildlife management units in Australia compared to the irregularly shaped polygons in NA and Europe, the latter of which is more familiar to me. Along those lines, the stark difference in time period for the Australian data is never mentioned, despite the fact that it serves to support the notion that the observed pattern is widespread while also raising concerns about data quality.
- 2) The GIS operations are described as if the reader were an ArcGIS user looking to click buttons and explore data. This is unacceptable given that ArcGIS is but one software program for doing GIS (with a very expensive license) and the operations are a mix of general and more detailed statistical functions being applied to spatial data. It is reasonable to make the reader aware of the software used during the analysis, but individual steps should be tied to the calculations being performed, not the specific tools in ArcGIS.
- 3) The description of piecemeal regression was lacking. This is not a common statistical approach and warrants more attention. For example, it is entirely unclear how supplemental table 1 is meant to be interpreted. Estimates of 0.00 with an SE of 0.00 for a regression coefficient?

This needs far more explanation.

4) The description of the spatial residual plots seems to be disconnected with what is being visually portrayed. As a scientist with expertise in spatial statistics, I would say that several of those plots suggest spatial autocorrelation, particularly for the data from Europe (both species) and Australia West (red fox). I am not going to suggest that the analyses presented here are flawed or that the authors need to use complex approaches to modeling this autocorrelation, but the implications should be given more attention. Residual plots often reveal interesting patterns and there is sometimes more to be learned about where a model does not fit, than where it does.

5) It is not clear to me why breakpoints would be expected, particularly for the predator. If there is a habitat gradient determining the predator abundance/distribution, that gradient could be gradual. The idea of a breakpoint for the mesopredator is more convincing, but even that does not seem necessary for the PEH to hold. Maybe I am missing/forgetting something here, but more explanation on why one should expect breakpoints would be helpful. The breakpoint suggests some biological mechanism is being triggered but it is not clear why that has to be case.

Specific comments:

L97: Should this be the start of the discussion? The narrative after this point does not correspond to a listing of uninterpreted facts, as most Results sections are often represented.

L147: suggests

L149-152: This is a really interesting and important observation.

L275: Can you briefly explain the cross validation method?

Figure 2. The scale of bounty values for dingoes vs. fox is far closer than those for wolf/fox and wolf/jackal. This figure is the first place that this is made obvious. Additional description of the data would be helpful.

Figure 4. This should be the first figure. It seems strange for the conceptual model of the hypothesis to be described last. Also, it is confusing for both breakpoints to be occurring at the same location – makes it seem as if this correspondence is part of the hypothesis. As I argued earlier, I am not convinced that any breakpoint is even necessary, but even if they are present, I have not seen any reasoning for why they would have to match in location. The data and analyses presented here suggest they often do NOT match.

Table S1: What are these “estimates”? The estimate of the intercept is obvious, but it unclear

what the other values represent. If this is a default table output from the R package used then that is great, but everything needs to be described properly.

Reviewer #2 (Remarks to the Author):

The objective of this work was to determine if there was an inverse relationship between the distribution of mesopredators and top predators. The reasoning for this suspected inverse relationship is well documented in the literature as there has been quite a bit of work on mesopredator suppression and release regarding the occurrence or lack of top predators in an area. So, as the authors point out, there is a lot of small scale data indicating a possible mechanism operating on a larger scale. Anecdotally, such suppression of mesopredators has been noted as early as 1959 by Stalker Leopold regarding the absence of coyotes in northern Mexico because of the presence of wolves (Leopold, A.S., 1959. Fauna Silvestre de Mexico, Editorial Pax, Mexico y Liberia Carlos Césarman, México, D.F.). Consequently, the idea to test this hypothesis on a large regional scale, which has not been done yet, is both relevant and original. Results of such a test would indeed have consequences regarding our developing knowledge of the role top predators play in ecosystems. Such results should be of interest to a wide variety of scientists as well as non-scientists.

Needless to say, however, to test this hypothesis on large landscape scales does present problems regarding other factors that might influence the distribution and abundance of both the mesopredator and top predators. However, I feel that the authors have adequately addressed these possible influences regarding their selection of study areas. Another problem with such large scale analyses is having accurate information on abundance of the test animals. Data on such large scales are rarely available. Again, the authors seem to have adequately addressed this regarding their use of bounty data. There indeed are weaknesses in using such data as a relative index of abundance but, as the authors pointed out, such data have been used in the past successfully for relative abundance and so should be adequate for the large scale trends they were testing for.

Consequently, I feel that the authors have compiled an adequate database to test their hypothesis. The fact that it encompasses three different continents indeed strengthens their analysis substantially. Their statistical analysis appears to be adequate.

Regarding the results, the fact that they did observe a similar pattern across the widely distributed study sites provides substantial support for their findings. I think that the conclusions of the authors are justified by the data.

John Laundré

Reviewer #3 (Remarks to the Author):

This paper uses predator bounty data as a proxy for predator abundances to analyze the impact of three top predators on three mesopredators (wolf-coyote in Saskatchewan, wolf-golden jackal in Bulgaria/Serbia, dingo-red fox in Queensland). In line with the results, the authors put forth the Predator Edge Hypothesis (PEH) suggesting that top-down suppression of mesopredators weakens towards the edge of the distribution of top predators as a result of declining top predator densities. They also point out that mesopredator densities often declined to levels close to zero within the range of the top predator, a result implied in the title of the paper. The authors conclude that the PEH can have conservation implications due to widespread and continued fragmentation of top predator ranges.

This is an interesting paper and the similarities between the study areas indeed suggest a general pattern. However, I think the paper needs some additions and clarifications, in particular relating to questions of originality and interpretations. I provide my concerns in detail below.

Originality. I think the overlap between this paper and Newsome and Ripple 2014 (in *Journal of Animal Ecology*) should be clearly stated. The authors also need to clarify what new knowledge this paper contributes in relation to previous findings. Although the 2014 paper include data from an additional region (Manitoba) and on one more species (fox, in addition to wolf and coyote), there appear to be substantial similarities in the results and conclusions regarding North America. One of the main results in 2014 was the transition zone of approximately 200 km from the distribution limit of the wolf, within which the top-down effect of wolf on coyote weakened. The addition in this new paper is that the finding is formalized (the PEH), and that similar results are found in two additional regions. I think this is interesting, but I basically find the results in the current paper confirmatory rather than new. Confirming previous findings is good and makes them more reliable, but I feel previous results must be clearly acknowledged.

Background theory. It is stated that little is known on the role of biotic interactions in determining range boundaries (line 43-45). I think this is worth expanding on, see e.g. the review by Louthan et al in *TREE*.

It is stated that “mesopredator abundance should vary with the spatial variation in the abundance of top predators” and “suppression of mesopredators should be greatest well within the edge of the top predator’s range where the abundances of that predator are highest” (line 51-56). I agree, and this pattern is seen clearly for dingo-fox in Australia west (Fig 3g, h). But I wonder about the

results for wolf-jackal (3c,d) and dingo-fox in Australia east (Fig. 3e,f). It appears as top predator densities first increase with increasing distance from their distribution edge, but then decline quite markedly. However, in the latter decline phase there is no corresponding increase in mesopredator densities. Could you provide an explanation for the pattern? Could there be confounding factors (ecological context, as mentioned in the abstract) which prevents mesopredators from increasing although top predator densities are low? Or is there some methodological explanation? I am also wondering about this because of the distribution pattern for wolf-jackal shown in Fig. 2b and e. It appears as if jackals are abundant primarily in a section of the study area, which happens to be within the "predator edge zone", but elsewhere jackals appears rare independent of whether wolf densities are high or low. So what additional factors are at play here?

Generality issue 1. It is stated several times in the paper (including the title) that top predator suppression can drive mesopredator densities to zero, or close to zero. Overall, I almost get the impression that the authors suggest that competitive exclusion of mesopredators is the expected ecosystem function of top predators. For example, they state that "for some distance within the edge of the top predator's range, suppression of mesopredators may occur but be insufficient to drive mesopredator abundances to zero" (line 56-58), "the mere presence of a top predator may not be sufficient to exert strong suppressive effects on mesopredators. This observation could explain why some studies have documented only weak effects of top predators on mesopredators" (line 149-152). This made me wonder, should not "weak effects", i.e. suppression rather than competitive exclusion, be the norm for many top predator-mesopredator interactions, at least within the native range of the mesopredators? Although there has been mesopredator release (in abundance and in some cases also distribution) following top predator declines, surely pristine ecosystems where top predators are omnipresent should still contain mesopredators? If the expected outcome of large-scale large carnivore restoration should be mesopredator extinction through competitive exclusion, would that not imply a risk to overestimate the ecosystem function of large carnivores? (compare to the author's phrasing in the conclusion on line 175-177). I suggest the authors should clarify under what circumstances one would expect suppression vs. competitive exclusion.

Generality issue 2. A related point. I note that the three study cases in this paper all concern mesopredators that have gone through relatively recent range expansions (coyote and jackal) or are invasive (fox in Australia). In the paper the authors state that all mesopredators in the study are wide-ranging generalists which have been studied in suitable habitat (line 99-105). However, in methods it is stated that the coyote has expanded its North American distribution substantially and Saskatchewan is not part of its historical distribution range. Likewise, jackal was historically only found in a small part of Bulgaria, but has expanded its distribution more recently. Red fox is introduced in Australia. I believe mesopredator release could be one reason why coyote and jackal have expanded, but then the study areas in this paper would perhaps not be within the

realized niche of the coyote and jackal? My point is, the three cases studied all concern mesopredators outside their historical/native range (or partially outside in the case of jackal). This means that they should not necessarily be able to co-exist with the top predators in question, at these study sites. So are the strong effects found in this paper, close to competitive exclusion, representative for top predators and mesopredators in ecosystems where they are native (in a longer historical context) and potentially have co-evolved? Again, when should competitive exclusion or suppression be the expected outcome of interactions between top predators and mesopredators? I think this could be clarified.

Reviewer Comment	Reply	Line numbers
Editor		
Please consider the following two points: 1) Considering making data and/or code available for evaluation by Reviewer 1. 2) Please explicitly acknowledge overlap between Newsome and Ripple (2015; JAE), giving credit to the prior paper in any instances where the same data are used. Please highlight all changes in the manuscript text file.	1) We have now made the code available in the supplementary material Table 4. 2) With respect to the overlap between Newsome and Ripple (2015; JAE), we have now cited this work explicitly in methods and we also note how this paper differs. In summary, Newsome and Ripple (2015; JAE) used a coyote-to-red fox ratio (calculated from bounty data) and compared changes in this ratio on either side of the grey wolf range. The key question addressed by Newsome and Ripple was whether a continental scale tri-trophic cascade existed across North America that extended from grey wolves through coyotes to red foxes. In the current paper, we ask a very different question from one site (Saskatchewan) considered by Newsome and Ripple. The main question concerns the relationship between grey wolves and coyotes and how that relationship changes across space. This question was not considered explicitly by Newsome and Ripple. Furthermore, Newsome and Ripple did not use grey wolf bounty data at all, and instead used a historical grey wolf range boundary to assess relationships between coyotes and red foxes on either side. Our current paper therefore uses data not considered by Newsome and Ripple. This is now stated in the methods section, and we have done the same for the other datasets examined (see lines 263-322). We hope this clarifies the concern.	263-322
Reviewer 1		
This paper examines the numerical suppression of mesopredator population abundance/distribution with increasing predator abundance using observed population gradients within predator ranges. The authors compiled data from 3 different predator systems across 3 different continents to illustrate a correspondence in the observed patterns, consistent with the proposed “predator edge hypothesis”. The paper is well written and the ideas are compelling. The figures are excellent and portray the story well.	Thanks for the positive feedback.	
My main criticism is that the data and methods are poorly described at times. I recognize that brevity is valued in a Nature Communications paper but in the absence of sharing data and code, there are more details needed to understand what was done. The authors provided a supplement of tables/figures and this supplement could be expanded with additional	Thanks, we have now added more detail to the methods and included two extra supplements. The supplement includes the r code used as well as a description of the bounty data (see Supplementary Tables 3 and 4). Updates in the methods carry from lines 263-388. We hope this helps to address the concern.	Supplementary Tables 3 & 4 & methods lines 263-388 (main body).

descriptions if such information cannot fit in the main document. Some general comments:		
1) The bounty data serve as the primary evidence for the hypothesis but these data are barely described in the text. Where exactly do these data come from? Citing a previous paper is not the same as citing a data source, and a citation alone is not sufficient for the reader to understand the data origin. Without any context, it is strange to see square wildlife management units in Australia compared to the irregularly shaped polygons in NA and Europe, the latter of which is more familiar to me. Along those lines, the stark difference in time period for the Australian data is never mentioned, despite the fact that it serves to support the notion that the observed pattern is widespread while also raising concerns about data quality.	Thanks, as mentioned above we have now added an extra supplement (see Supplementary Table 3) that describes the bounty data in more detail. With respect to the time span of the bounty data, we have added a sentence to the discussion to explicitly state that the data from Australia is from a much older time series than in North America and Europe, but that it nonetheless is still relevant to consider (see lines 141-144)	Supplementary Table 3 and lines 141-144 (main body)
2) The GIS operations are described as if the reader were an ArcGIS user looking to click buttons and explore data. This is unacceptable given that ArcGIS is but one software program for doing GIS (with a very expensive license) and the operations are a mix of general and more detailed statistical functions being applied to spatial data. It is reasonable to make the reader aware of the software used during the analysis, but individual steps should be tied to the calculations being performed, not the specific tools in ArcGIS.	Thanks for picking this up. We have now added more detail to the methods to clarify the individual steps and methods used (see lines 343-349; 352-355; 373-378)	Lines 343-349; 352-355; 373-378.
3) The description of piecemeal regression was lacking. This is not a common statistical approach and warrants more attention. For example, it is entirely unclear how supplemental table 1 is meant to be interpreted. Estimates of 0.00 with an SE of 0.00 for a regression coefficient? This needs far more explanation.	To address these concerns we have added additional text to the methods (lines 373- 378) as well as to supplemental table 1. We have also added the code to the supplement which should aid interpretations (Supplementary Table 4). For supplemental table 1, we presented the data at the two decimal place level, which results in estimates and SE of 0.00, but we have now made these to the five decimal point level, or the nearest integer, to avoid confusion.	Lines 373-378; Supplementary Table 4)
4) The description of the spatial residual plots seems to be disconnected with what is being visually portrayed. As a scientist with expertise in spatial statistics, I would say that several of those plots suggest	This is a valid point, so we have now added additional text to the results to acknowledge the spatial clustering of the residuals in Europe (both species) and Australia West (red fox). In particular, we have noted that there is some minor	Lines 100-102

spatial autocorrelation, particularly for the data from Europe (both species) and Australia West (red fox). I am not going to suggest that the analyses presented here are flawed or that the authors need to use complex approaches to modeling this autocorrelation, but the implications should be given more attention. Residual plots often reveal interesting patterns and there is sometimes more to be learned about where a model does not fit, than where it does.	clustering in the residuals (lines 100-102) but this clustering is not in any particular direction that is of relevance to the results found, so we do not believe this warrants any further investigation.	
5) It is not clear to me why breakpoints would be expected, particularly for the predator. If there is a habitat gradient determining the predator abundance/distribution, that gradient could be gradual. The idea of a breakpoint for the mesopredator is more convincing, but even that does not seem necessary for the PEH to hold. Maybe I am missing/forgetting something here, but more explanation on why one should expect breakpoints would be helpful. The breakpoint suggests some biological mechanism is being triggered but it is not clear why that has to be case.	This is a valid point, and we agree that more detail should be provided to clarify why the breakpoints are included. To that end, we have added text to Figure 1 (previously Figure 4) to clarify why and where a breakpoint could occur, but that the PEH could hold even if the breakpoint does not exist. Furthermore, it is worth noting that we used the piecewise regression not so much because we expected breakpoints but rather because this approach discriminates between scenarios where relationships are described by one slope versus several slopes. This, we feel, was important because the large spatial scale of our analysis made it more likely that slopes describing top down effects would vary. As further background, the original reason for the breakpoint was that we expected there to be an “ecological effective density” of the top predator to be reached before the suppression of mesopredators could occur/be detected. Our results lend support to this, and we note this in the discussion (lines 150-154).	Figure 1 (previously Figure 4), lines 216-226; lines 150-154
L97: Should this be the start of the discussion? The narrative after this point does not correspond to a listing of uninterpreted facts, as most Results sections are often represented.	Thanks, we have now moved the headings around to overcome this problem.	Line 103
L147: suggests	Change adopted.	Line 159
L149-152: This is a really interesting and important observation.	Thanks.	
L275: Can you briefly explain the cross validation method?	Additional text has now been added to the methods. In short, the cross validation method is part of the smoothing process for the kernel density estimate. It was chosen because it does not typically overestimate space use, and thus provides a conservative kernel.	Lines 347-349
Figure 2. The scale of bounty values for dingoes vs. fox is far closer than those for wolf/fox and wolf/jackal. This figure is the first place that this is made obvious. Additional description of the data would be helpful.	Thanks, the new supplement Table 3 contains extra detail about the bounty data, and we have added new text to the Figure captions (Figure 1 and 2) to ensure readers are aware of the differences in scales (see lines 229 and 235). We had to modify the scale so that each of the figures could be easily seen and compared.	Lines 229 & 235

Figure 4. This should be the first figure. It seems strange for the conceptual model of the hypothesis to be described last. Also, it is confusing for both breakpoints to be occurring at the same location – makes it seem as if this correspondence is part of the hypothesis. As I argued earlier, I am not convinced that any breakpoint is even necessary, but even if they are present, I have not seen any reasoning for why they would have to match in location. The data and analyses presented here suggest they often do NOT match.	This is a good suggestion and so we have now changed the figure order around to make Figure 4 now Figure 1. For the second point, we have acknowledged in the discussion that the breakpoints for North America and Australia West do not directly align and discussed the implication (lines 164-177). Furthermore, we agree that breakpoints do not necessarily need to occur at the same location or even be present for the PEH to hold. To help stress this point we have updated Figure 1 (previously Figure 4, lines 216-226) and in lines 150-157 we have stated that the existence of the breakpoints are not essential but that they may identify abundance thresholds where the top predator becomes ecologically effective and is therefore useful to assess. We hope the new text helps to clarify these issues.	Lines 164-177; 216-226; 150-157.
Table S1: What are these “estimates”? The estimate of the intercept is obvious, but it unclear what the other values represent. If this is a default table output from the R package used then that is great, but everything needs to be described properly.	We have added text to make sure this is clear in Supplementary Table 1. In particular we state that “Estimates for the intercept (Intercept) and slope of first line segment (Line 1 85%) are shown with standard errors (SE). Numbers for line 2 85% represent the difference in slopes between the first and second line segment within each regression (see Fig. 3). Line 2 is significant ($P < 0.05$) when the slope of line 2 differs from the slope of line 1 at the breakpoint”	Supplementary Table 1
Reviewer 2		
The objective of this work was to determine if there was an inverse relationship between the distribution of mesopredators and top predators. The reasoning for this suspected inverse relationship is well documented in the literature as there has been quite a bit of work on mesopredator suppression and release regarding the occurrence or lack of top predators in an area. So, as the authors point out, there is a lot of small scale data indicating a possible mechanism operating on a larger scale. Anecdotally, such suppression of mesopredators has been noted as early as 1959 by Stalker Leopold regarding the absence of coyotes in northern Mexico because of the presence of wolves (Leopold, A.S., 1959. Fauna Silvestre de Mexico, Editorial Pax, Mexico y Liberia Carlos Césarman, México, D.F.). Consequently, the idea to test this hypothesis on a large regional scale, which has not been done yet, is both relevant and original. Results of such a test would indeed	Thank you for the positive and constructive feedback.	

have consequences regarding our developing knowledge of the role top predators play in ecosystems. Such results should be of interest to a wide variety of scientists as well as non-scientists.		
Needless to say, however, to test this hypothesis on large landscape scales does present problems regarding other factors that might influence the distribution and abundance of both the mesopredator and top predators. However, I feel that the authors have adequately addressed these possible influences regarding their selection of study areas. Another problem with such large scale analyses is having accurate information on abundance of the test animals. Data on such large scales are rarely available. Again, the authors seem to have adequately addressed this regarding their use of bounty data. There indeed are weaknesses in using such data as a relative index of abundance but, as the authors pointed out, such data have been used in the past successfully for relative abundance and so should be adequate for the large scale trends they were testing for.	Thank you for the constructive feedback. We agree there are factors that we cannot test, and that bounty data have limitations. In the absence of any other data, however, we strongly believe that bounty data can be used to test broad spatial trends to test specific hypotheses. Furthermore, in our paper we have encouraged further testing of the PEH, and we hope this eventuates.	
Consequently, I feel that the authors have compiled an adequate database to test their hypothesis. The fact that it encompasses three different continents indeed strengthen their analysis substantially. Their statistical analysis appears to be adequate.	Thanks.	
Regarding the results, the fact that they did observe a similar pattern across the widely distributed study sites provides substantial support for their findings. I think that the conclusions of the authors are justified by the data.	Thanks.	
Reviewer 3		
This paper uses predator bounty data as a proxy for predator abundances to analyze the impact of three top predators on three mesopredators (wolf-coyote in Saskatchewan, wolf-golden jackal in Bulgaria/Serbia, dingo-red fox in Queensland). In line with the results, the authors put forth the Predator Edge Hypothesis (PEH) suggesting that top-down suppression of mesopredators weakens towards the edge of the distribution of top	Thank you for the positive and constructive feedback.	

predators as a result of declining top predator densities. They also point out that mesopredator densities often declined to levels close to zero within the range of the top predator, a result implied in the title of the paper. The authors conclude that the PEH can have conservation implications due to widespread and continued fragmentation of top predator ranges.		
This is an interesting paper and the similarities between the study areas indeed suggest a general pattern. However, I think the paper needs some additions and clarifications, in particular relating to questions of originality and interpretations. I provide my concerns in detail below.	Thanks. Based on this comment and the other reviewers we have added a substantial amount of new text as well as two new supplements to help clarify the methods and interpretation. We hope that the additional text and supplements provided help to address your concerns.	See red/blue text in document.
Originality. I think the overlap between this paper and Newsome and Ripple 2014 (in Journal of Animal Ecology) should be clearly stated. The authors also need to clarify what new knowledge this paper contributes in relation to previous findings. Although the 2014 paper include data from an additional region (Manitoba) and on one more species (fox, in addition to wolf and coyote), there appear to be substantial similarities in the results and conclusions regarding North America. One of the main results in 2014 was the transition zone of approximately 200 km from the distribution limit of the wolf, within which the top-down effect of wolf on coyote weakened. The addition in this new paper is that the finding is formalized (the PEH), and that similar results are found in two additional regions. I think this is interesting, but I basically find the results in the current paper confirmatory rather than new. Confirming previous findings is good and makes them more reliable, but I feel previous results must be clearly acknowledged.	Thanks, as noted above, we have now added more detail on how our analysis differs from what was presented in Newsome and Ripple 2014. In particular, as we note above: With respect to the overlap between Newsome and Ripple (2015; JAE), we have now cited this work explicitly in methods and we also note how this paper differs. In summary, Newsome and Ripple (2015; JAE) used a coyote-to-red fox ratio (calculated from bounty data) and compared changes in this ratio on either side of the grey wolf range. The key question addressed by Newsome and Ripple was whether a continental scale tri-trophic cascade existed across North America that extended from grey wolves through coyotes to red foxes. In the current paper, we ask a very different question from one site (Saskatchewan) considered by Newsome and Ripple. The main question concerns the relationship between grey wolves and coyotes and how that relationship changes across space. This question was not considered explicitly by Newsome and Ripple. Furthermore, Newsome and Ripple did not use grey wolf bounty data at all, and instead used a historical grey wolf range boundary to assess relationships between coyotes and red foxes on either side. Our current paper therefore uses data not considered by Newsome and Ripple. This is now stated in the methods section, and we have done the same for the other datasets examined (see lines 263-322).	Lines 263-322
Background theory. It is stated that little is known on the role of biotic interactions in determining range boundaries (line 43-45). I think this is worth expanding on, see e.g. the review by Louthan et al in TREE.	Thanks, we have included related text and cited the Louthan et al. paper in the introduction and discussion. In particular, Louthan et al focus on the role of abiotic stressors, so we have also given additional attention to this point.	Line 48 & 171
It is stated that “mesopredator abundance should vary with the spatial variation in the abundance of top	This is a valid point – and we think this may be linked to habitat suitability in both systems.	Lines 168-177

predators” and “suppression of mesopredators should be greatest well within the edge of the top predator’s range where the abundances of that predator are highest” (line 51-56). I agree, and this pattern is seen clearly for dingo-fox in Australia west (Fig 3g, h). But I wonder about the results for wolf-jackal (3c,d) and dingo-fox in Australia east (Fig. 3e,f). It appears as top predator densities first increase with increasing distance from their distribution edge, but then decline quite markedly. However, in the latter decline phase there is no corresponding increase in mesopredator densities. Could you provide an explanation for the pattern? Could there be confounding factors (ecological context, as mentioned in the abstract) which prevents mesopredators from increasing although top predator densities are low? Or is there some methodological explanation? I am also wondering about this because of the distribution pattern for wolf-jackal shown in Fig. 2b and e. It appears as if jackals are abundant primarily in a section of the study area, which happens to be within the “predator edge zone”, but elsewhere jackals appears rare independent of whether wolf densities are high or low. So what additional factors are at play here?	For example, in Australia East the top predator densities decline in an area where population density of humans is high (the east coast) and in Europe both jackals and grey wolves are virtually absent from northern Serbia where there is intensive agriculture. We have now added a section to clarify this point (lines 168-177)	
Generality issue 1. It is stated several times in the paper (including the title) that top predator suppression can drive mesopredator densities to zero, or close to zero. Overall, I almost get the impression that the authors suggest that competitive exclusion of mesopredators is the expected ecosystem function of top predators. For example, they state that "for some distance within the edge of the top predator’s range, suppression of mesopredators may occur but be insufficient to drive mesopredator abundances to zero" (line 56-58), "the mere presence of a top predator may not be sufficient to exert strong suppressive effects on mesopredators. This observation could explain why some studies have documented only weak effects of top predators on mesopredators" (line 149-152). This made me wonder, should not "weak	This is a very good point and we agree. However, we don’t think we can identify from our data the exact situations where complete exclusion would occur as opposed to suppression. Therefore, we have suggested that this becomes a focus of future studies. In doing so, however, we do note that complete exclusion of mesopredators occurred from vast areas when top predators were at historical levels. We also provide an example where complete exclusion and coexistence has occurred more recently (lines 186-196). As a side note, testing for mesopredator suppression in general has been difficult, and contemporary support would require large-scale field experiments. As a starting point however, studies like ours can help inform general patterns and are therefore useful to address this challenge.	Lines 186-196

effects", i.e. suppression rather than competitive exclusion, be the norm for many top predator-mesopredator interactions, at least within the native range of the mesopredators? Although there has been mesopredator release (in abundance and in some cases also distribution) following top predator declines, surely pristine ecosystems where top predators are omnipresent should still contain mesopredators? If the expected outcome of large-scale large carnivore restoration should be mesopredator extinction through competitive exclusion, would that not imply a risk to overestimate the ecosystem function of large carnivores? (compare to the author's phrasing in the conclusion on line 175-177). I suggest the authors should clarify under what circumstances one would expect suppression vs. competitive exclusion.		
Generality issue 2. A related point. I note that the three study cases in this paper all concern mesopredators that have gone through relatively recent range expansions (coyote and jackal) or are invasive (fox in Australia). In the paper the authors state that all mesopredators in the study are wide-ranging generalists which have been studied in suitable habitat (line 99-105). However, in methods it is stated that the coyote has expanded its North American distribution substantially and Saskatchewan is not part of its historical distribution range. Likewise, jackal was historically only found in a small part of Bulgaria, but has expanded its distribution more recently. Red fox is introduced in Australia. I believe mesopredator release could be one reason why coyote and jackal have expanded, but then the study areas in this paper would perhaps not be within the realized niche of the coyote and jackal? My point is, the three cases studied all concern mesopredators outside their historical/native range (or partially outside in the case of jackal). This means that they should not necessarily be able to co-exist with the top predators in question, at these study sites. So are the strong effects found in this paper, close to	Thanks, as noted above, we have now added additional text to the discussion on the topic of complete exclusion versus suppression (lines 186-196). Our paper suggests complete exclusion may be present in the core of the top predator ranges, but we also note that further testing would be required to fully determine if this is the case or an artefact of low detection (lines 193-196). With respect to the question of co-evolution, all three of the mesopredators coexist with their respective top predators within parts of the current range (and in some instances their historical range), including across a range of different habitats (see Figure 2). For this reason, we think the key question is not whether these predators can coexist, but whether mesopredator abundance is affected by top predator abundance in areas where they have overlapping ranges (the key focus of the paper).	Lines 186-196

competitive exclusion, representative for top predators and mesopredators in ecosystems where they are native (in a longer historical context) and potentially have co-evolved? Again, when should competitive exclusion or suppression be the expected outcome of interactions between top predators and mesopredators? I think this could be clarified.

Reviewers' Comments:

Reviewer #1 (Remarks to the Author):

Thank you for addressing the relevant concerns.

Reviewer #3 (Remarks to the Author):

I thank the authors for their clear responses to my comments, which all have been taken into account and the paper has been revised accordingly. I have no further comments and would like to congratulate the authors on their interesting study.

However, I would like to mention a thought I had when reading the new version of the paper and the authors' responses. In line 197-198, the authors state that "the general predictions of PEH can be tested for other predator dyads that compete for similar resources". Should this theory not have the potential to be even more general? I understand that the authors confine the hypothesis to predators since that is the species dyad this study concerns, but should it not be applicable also to other species dyads? If so, the hypothesis could be extended to a "Species/Competition/Distribution Edge Hypothesis" which might be applicable to all species dyads where biotic competitive factors determine species distributions. Perhaps this potential extension could be worth mentioning somewhere around line 197-198? As a conclusion, it would relate back quite nicely to the introductory sentences in line 43-36.

Reviewer Comment	Reply	Line numbers (with markup shown)
Editor		
When you submit your revised manuscript, please submit a word doc with "track changes" function on, so that I can easily identify the changes made. Please make sure that all figures and tables, main and supplementary, are referred to at least once in the manuscript. Also, please make sure that the first referral to any given figure is in numerical order (i.e. you should not be referring to Fig. 2 before you mention Fig. 1, or to Supplementary Fig. 2 before Supplementary Fig. 1).	All changes made are in track changes. References to the figures and supplementary material are in the correct order.	Lines 1-517
DRAFT EDITOR'S SUMMARY (please approve/edit but maintain ~325 character limit, including spaces):	Editor's summary is included below the title; some minor changes have been suggested.	Lines 3-7
Please consider a slightly expanded title	Thanks, but we would like to retain the shorter title because "distribution" implies that we are already assessing the relationships between top predators and mesopredators at a large spatial scale.	Line 1
Please provide postal codes for all affiliations.	Completed.	Lines 13-30
Please shorten the abstract to 150 words.	Abstract has been shortened, it is now 150 words	Lines 31-46
The Introduction must end with a paragraph that briefly summarizes the main results and conclusions.	We have now added a new paragraph to the end of the introduction to meet this requirement.	Lines 88-98
The Results text must be structured into discrete subsections with subheadings of 60 characters or less (including spaces).	Three new subheadings have been added.	Lines 100-120
Please address Reviewer 3's point by commenting on the possibility of generalizing the hypothesis to other types of ecological interactions.	In order to meet this request (which we agree with) we have taken on board the reviewer's suggestion to broaden the description of the hypothesis. This first required a new name for our hypothesis as the "Predator Edge Hypothesis" only relates to predators. Therefore, we have changed our hypothesis to the "Enemy Constraint Hypothesis" (ECH). In doing so, there is no change to the underlying assumptions of the hypothesis, but it allows us to make the call for future studies to test whether the results of the ECH apply to other predator dyads as well as to any strongly interacting competitive species dyads. This change has been made throughout the paper, and the text has been modified in Figure 1 to ensure clarity.	Lines 1-517

Please indicate in the next cover letter whether the images of the animals are original (i.e. drawn by you) or if they are stock images. If latter, you will need to obtain third party image permission to reproduce them in your paper.	The cover letter contains all this information.	n/a
Please delete figures from the main text and upload as individual files in the final submission.	Figures have been deleted.	n/a
See note above about the third party images/photographs used here.	The cover letter contains all this information.	n/a
Please describe each panel individually. Also, please describe in text what is denoted by the black box and dot shapes, the hash marks, and yellow color. Is the dashed line in Panel A (separating US and Canada) necessary? There are not comparable distinctions made on panel B, for example, so I wonder if that can be dropped.	The legend for Figure 2 is now updated, with each panel described individually. In Figure 3 the dashed line separating US and Canada has been removed.	Lines 397-417
Please describe the nature of the color scale (i.e. – darker red colors indicate greater hunting bounties). Also, what is the unit on bounties? Number of animals?	The legend for Figure 2 is now updated to include this information.	Lines 397-403
Please consider making these symbols larger – they are hard to spot.	Symbols have been made larger.	n/a
All Nature Communications manuscripts must include a Data availability statement at the end of the Methods section or main text (if no Methods). For more information on this policy, and a list of examples, please see http://www.nature.com/authors/policies/data/data-availability-statements-data-citations.pdf In particular, the Data availability statement should include:  - Accession codes for deposited data - Other unique identifiers (such as DOIs and hyperlinks for any other datasets) - At a minimum, a statement confirming that all relevant data are available from the authors - If applicable, a statement regarding data available with restrictions - If a dataset has a Digital Object Identifier (DOI) as its unique identifier, we strongly encourage including this in the Reference list and citing the dataset in the Data Availability Statement. 	Data availability statement has been added, and the data will be uploaded to Dryad upon final acceptance.	Lines 379-381
Acknowledgements Author Contributions Competing financial interests statement	Information added.	Lines 519-528

Reviewer 3		
I thank the authors for their clear responses to my comments, which all have been taken into account and the paper has been revised accordingly. I have no further comments and would like to congratulate the authors on their interesting study.	As noted above, we have taken on board this suggestion and made the hypothesis more general. In doing so, we have changed the name of the hypothesis to the “Enemy Constraint Hypothesis” (ECH). This name was chosen over the suggestions by the reviewer (“Species/Competition/Distribution Edge Hypothesis”) because the word “enemy” has a long history of use covering predators, competitors, pathogens, parasites etc, at least going back to Elton’s work on invasions (invader populations expand rapidly in the absence of ‘natural enemies’). A more recent use is in the concept of ‘enemy-free space’, where individual fitness increases for organisms where their enemies do not occur (Jeffries and Lawton 1984: Biol J Linn Soc 23). The word “Constraint” is used because it can apply to a reduction or limit to a species’ abundance and fitness, both of which are linked to the effects of predation and competition. Finally, as far as we are aware, the acronym ECH has not been used previously in our field.	Line 39, 73, 77, 92, 96, 168, 215-218, 220, 371, 382-394.
However, I would like to mention a thought I had when reading the new version of the paper and the authors’ responses. In line 197-198, the authors state that “the general predictions of PEH can be tested for other predator dyads that compete for similar resources”. Should this theory not have the potential to be even more general? I understand that the authors confine the hypothesis to predators since that is the species dyad this study concerns, but should it not be applicable also to other species dyads? If so, the hypothesis could be extended to a “Species/Competition/Distribution Edge Hypothesis” which might be applicable to all species dyads where biotic competitive factors determine species distributions. Perhaps this potential extension could be worth mentioning somewhere around line 197-198? As a conclusion, it would relate back quite nicely to the introductory sentences in line 43-36.	The broader applicability of this hypothesis is now mentioned at the end of the introduction as well as in Fig. 1.